# Combined Pulmonary Fibrosis and Emphysema: Comparative Evidence on a Complex Condition

**DOI:** 10.3390/biomedicines11061636

**Published:** 2023-06-04

**Authors:** Diana Calaras, Alexander G. Mathioudakis, Zsofia Lazar, Alexandru Corlateanu

**Affiliations:** 1Department of Pulmonology and Allergology, State University of Medicine and Pharmacy “Nicolae Testemitanu”, MD-2004 Chisinau, Moldova; alexandru.corlateanu@usmf.md; 2Division of Immunology, Immunity to Infection and Respiratory Medicine, School of Biological Sciences, The University of Manchester, Manchester M13 9PL, UK; 3Department of Pulmonology, Semmelweis University, 1083 Budapest, Hungary

**Keywords:** emphysema, lung fibrosis, interstitial lung disease, lung function, pulmonary hypertension

## Abstract

Combined pulmonary fibrosis and emphysema (CPFE) is a clinical syndrome characterized by upper lobe emphysema and lower lobe fibrosis manifested by exercise hypoxemia, normal lung volumes, and severe reduction of diffusion capacity of carbon monoxide. It has varying prevalence worldwide with a male predominance, and with smoking history of more than 40 pack-years being a common risk factor. The unique imaging features of CPFE emphasize its distinct entity, aiding in the timely detection of pulmonary hypertension and lung cancer, both of which are common complications. High-resolution computed tomography (HRCT) is an important diagnostic and prognostic tool, while lung cancer is an independent factor that alters the prognosis in CPFE patients. Treatment options for CPFE are limited, but smoking cessation, usual treatments of pulmonary fibrosis and emphysema, and avoidance of environmental exposures are encouraged.

## 1. Introduction

Chronic obstructive pulmonary disease (COPD) and idiopathic pulmonary fibrosis (IPF) have long been considered mutually exclusive disorders. However, in recent years, the coexistence of emphysema and pulmonary fibrosis has emerged as a new entity. Historically, these two diseases found concomitantly in the same patient were first described almost half a century ago [1], and it was only in 2005 that Cottin and colleagues proposed the term “combined pulmonary fibrosis and emphysema” (CPFE) as the name of a well-defined condition characterized by upper lobe emphysema and lower lobe pulmonary fibrosis [2], to mark a specific consequence of smoking in susceptible patients. CPFE is considered to have distinct pathophysiological, clinical, radiological, functional, and prognostic features compared to its separate components.

In light of the increased interest in this new entity over the past few years, this article will present a comprehensive overview of the specific features of CPFE and comparative evidence describing COPD and IPF. 

## 2. Definition

CPFE is a clinical syndrome that affects heavy smokers and is characterized by a combination of upper lobe emphysema and lower lobe fibrosis. Symptoms of CPFE include exercise-induced and later resting hypoxemia, normal lung volumes, and a significant reduction in the diffusing capacity, frequently associated with pulmonary hypertension. It is a heterogeneous condition affecting a diverse population of individuals.

## 3. Epidemiology

COPD prevalence estimates vary across the world in the general population, ranging from 1382.6 cases per 100,000 in Latin America and 1821.5 cases per 100,000 in Eastern Europe to 3017.5 cases per 100,000 in Western Europe and 3558.4 cases per 100,000 in North America [3]. In contrast, idiopathic pulmonary fibrosis (IPF) has a lower prevalence, ranging from 0.33 to 2.51 cases per 10,000 people in Europe, 2.40 to 2.98 in North America, and 0.57 to 4.51 in Asia-Pacific countries [4]. CPFE is estimated to occur in 8–67% of IPF patients, with variations in prevalence depending on the population studied, genetic susceptibility, smoking rates, or CPFE definitions [2,5,6]. Conversely, lung fibrosis was found to occur in 8% of emphysema patients based on HRCT scans [7].

Patients with CPFE are generally older with a heavy smoking history. The majority of studies have shown that almost all CPFE patients have a significant smoking history, suggesting that smoking may be the predominant risk factor for this condition [8,9]. The smoking history does not differ considerably between CPFE and COPD [10], although IPF patients often have fewer pack-years than those with CPFE or COPD [11].

Numerous studies have found a male predominance in CPFE syndrome [12], but more recent data [8] indicate a significant female incidence. Although it has been shown that male smokers are more likely than female smokers to develop emphysema and IPF [13], this does not necessarily mean that gender is a separate risk factor for CPFE. To fully understand the gender variations in this disease, more research is required.

After the sixth decade of life, CPFE seems to affect males more frequently [14]. Patients typically have a smoking history of more than 40 pack-years and are either current or former smokers [15]. However, studies showing CPFE in non-smokers with connective tissue disease have demonstrated a genetic predisposition to the disease [16].

## 4. Etiology and Pathogenesis

The exact mechanisms leading to the development of both emphysema and fibrotic lesions are not fully understood. Whether the coexistence of two different entities is just a “cohabitation” or if there is a result of shared mechanisms that led to the creation of both is still up for debate. However, the pathogenesis of CPFE is multifactorial, occurring in susceptible individuals after exposure to environmental triggers such as smoking or dust inhalation. 

Emphysema and pulmonary fibrosis are both recognized as primarily driven by smoking [17], and most CPFE patients appear to be either current or former smokers [15]. The correlation between CPFE and pack years supports a dose–response relationship [18,19]. 

Environmental exposures such as noxious particles or gases other than tobacco, asbestos, silica dust, and agrochemical chemicals [20] may contribute to lung damage in CPFE syndrome patients, even in lifelong nonsmokers. Moreover, case-based reports link welding jobs or the tire manufacturing industry to CPFE [21,22]. Interestingly, CPFE can be found in up to 23% of patients with fibrotic hypersensitivity pneumonitis (fHP) [23]. 

Since nonsmokers also develop CPFE, which is especially true in connective tissue diseases (CTDs) [24,25,26], the condition itself could be considered a risk factor. Radiological findings of CPFE have been documented in systemic sclerosis-associated ILD [25], rheumatoid arthritis-associated ILD [24], systemic vasculitis-associated ILD, and particularly microscopic polyangiitis [26]. Regarding autoimmunity as a potential contributing background, compared to IPF patients, individuals with CPFE were likelier to have high serum antinuclear antibodies, with or without positive perinuclear antineutrophil cytoplasmic antibodies (p-ANCA), and also a better prognosis [27]. The better outcomes in patients with CTD-associated CPFE compared to those with idiopathic CPFE could be linked to an increased infiltration of CD20+ B lymphocytes generating lymphoid follicles in fibrotic lung tissue [27].

Another potential risk factor for the development of both entities is oxidative stress [28,29]. In the case of emphysema, numerous hazardous substances found in cigarette smoke alter bronchoalveolar macrophages, producing an excess of reactive oxygen species (ROS) and nitrogen species (RNS). This results in the production of pro-inflammatory cytokines and the recruitment of inflammatory cells, which generate ROS and increases the oxidative stress burden, impairing the balance of proteases and antiproteases, leading to DNA damage and causing cellular injury and death and decrease in extracellular matrix proteases [29,30]. However, in the case of pulmonary fibrosis, the same oxidizing agents (cigarette smoke, hyperoxia, asbestos, drugs and radiation) induce the production of ROS/RNS, stimulate the production of pro-fibrotic factors—transforming growth factor β (TGF-β)—that activate fibroblasts and their differentiation into myofibroblasts, as well as extracellular matrix deposition [31].

In addition, accelerated lung aging has been suggested as a potential mechanism for the onset of pulmonary fibrosis and pulmonary emphysema [17,32], and both disorders have been linked to senescence markers and mutations in genes associated with lung surfactant [33]. Telomere-shortening mutations in genes encoding telomerase are risk factors for pulmonary fibrosis in up to 20% of familial cases [34]. Similarly, short telomeres can lower the threshold of smoking-induced damage and constitute a genetic susceptibility factor for emphysema.

It is reasonable to assume that the lung parenchyma depicts distinct patterns of injury and repair, displaying various phenotypes of lesions determined by the balance between apoptosis, proteolysis, and fibrosis. Patients with overexpression of genes related to connective tissue synthesis, structural components of the cytoskeleton, and cell adhesion typically exhibit a fibrogenic phenotype similar to that observed in patients with IPF; however, a different inflammatory response to smoking-associated cellular damage (destruction and repair of cells, vessels, and pneumocytes) results in the destruction of the lung parenchyma, leading to pulmonary emphysema [35]. Gene expression analysis of fibrotic and emphysematous lesions in patients with CPFE has demonstrated the presence of a combination of fibrogenic and emphysematous patterns, reflecting ongoing damage to alveolar epithelial cells, attempts at alveolar regeneration, uncontrolled fibrosis proliferation, and parenchymal destruction, resulting in a vicious cycle of continuous injury and fibrosis [36].

Studies suggest that the development of CPFE results from the interaction between environmental exposure, such as smoking, and genetic predisposition in susceptible individuals. COPD and IPF share genetic variants in the MMP9, MUC5B, FAM13A, DSP, and TERT genes [37,38]. Since each disease has a unique pathophysiological profile, it is unknown whether the interaction between COPD and IPF can lead to the development of CPFE. Intriguingly, findings suggest that COPD, IPF, and CPFE syndrome each have a distinct genomic profile. Moreover, studies found that the same allele may have opposite roles in these two conditions. For example, a FAM13A allele was associated with an increased risk of IPF and a decreased risk of COPD [39]. Others report an association between either COPD or IPF with an increased risk of CPFE. A study performed in a Japanese cohort reported that the minor allele of the AGER gene rs2070600 was associated with CPFE among COPD patients [40]. Earlier, it was found that the T allele of MMP9 may be a risk factor for developing emphysema in patients with IPF in the Chinese population [41]. Moreover, the rs2736100 C allele of TERT was associated with a decreased IPF risk and an increased risk for CPFE in a Mexican cohort. Additionally, the rs2076295 TT genotype of DSP was associated with an increased IPF risk, while the GG genotype with CPFE susceptibility [42]. In a recent study, Ghosh, A.J. and colleagues showed divergent gene expression profiles in COPD and IPF, emphasizing opposing inflammatory and immune-related pathways in the pathogenesis of both diseases. Nonetheless, the overexpression of this gene signature in the blood was associated with decreased lung function in both diseases, indicating the presence of a common inflammatory subtype associated with a more severe disease course [43]. 

Despite sharing age-related alterations, mitochondrial dysfunction, excessive oxidative stress, resulting in pathological tissue repair, the prevalence, pathology, and clinical behavior of IPF and COPD are notably distinct. This is likely due to substantial disparities in genetic background and epigenetic modifications between the two diseases, which result in distinct target cell types and molecular responses to environmental triggers. As such, the mechanisms that contribute to the age-related preference for IPF and COPD, or the combination, currently remain unknown [44].

A summary of the factors associated with CPFE is represented in Figure 1. 

## 5. Main Pathological Features

COPD and IPF share common features at both the micro- and macroscopic levels, demonstrating patchy distribution of pathological areas, consisting of extracellular matrix proteins or inflammatory infiltrates with fibrotic regions alternating with regions of normal alveolar tissue or areas with interstitial leukocyte accumulation [45]. The distinction between the two entities consists mainly in the affected compartments of the lung, small airway remodeling, fibrosis, and destruction of the lung parenchyma with an upper lobe predominance in COPD while IPF exclusively affects the lung parenchyma and interstitium, having a lower lobe predilection [46]. Lung alterations found in CPFE most commonly combine emphysema and patchy fibrosis, fibroblast foci, and honeycombing comprising the usual interstitial pneumonia (UIP) pattern. A hallmark of CPFE is the presence of thick-walled cysts on a UIP background, which is a combination of emphysema and smoking-related interstitial fibrosis (SRIF). Alternatively, it may associate the nonspecific interstitial pneumonitis (NSIP) or the desquamative interstitial pneumonitis (DIP) patterns [6].

## 6. Clinical Features

Compared to IPF and COPD, patients with CPFE typically have a mean age of 65–70 years, and males predominantly account for 73–100% of cases [6]. 

Cough and dyspnea are frequent symptoms in CPFE, COPD, and IPF. Typically, chronic cough and sputum production, as a hallmark of COPD, occur many years before airflow limitation [47]. In contrast, patients with IPF experience dyspnea in over 90% of cases at the time of diagnosis [48], followed by a dry cough in 80% of patients in the late stage [49]. The symptoms of CPFE resemble those of IPF more closely. 

In almost all patients, progressive shortness of breath is the most common symptom and is typically more severe, especially during physical exertion. Exertional dyspnea is the clinical expression of inefficient ventilation and increased dead space ventilation in hypoperfused areas [50]. 

Other common respiratory signs and symptoms, such as cough, wheezing, cyanosis, and fatigue, may also manifest in some patients. 

On physical examination, patients with CPFE typically have inspiratory crackles referred to as “velcro sounds” produced by the underlying pulmonary fibrosis, found in 87–100% of patients, and almost half of them have finger clubbing [15]. In addition, pulmonary hypertension (PH) imposes a New York Heart Association functional class of III or IV during physical exertion [51] and may contribute to peripheral edema and hepatosplenomegaly. Lung cancer and PH are the two most prevalent comorbidities in CPFE [52,53].

## 7. Lung Function

CPFE is distinguished from pure emphysema and IPF in terms of pulmonary function by the unexpected presence of relatively normal lung volumes in contrast to a severely diminished diffusing capacity (DLCO), as presented in Table 1 [9]. The preserved lung volumes can be attributed to the counterbalancing effects of emphysema’s hyperinflation defect and pulmonary fibrosis’ restrictive defect. Thus, Cottin et al. concluded that serial FVC measurement may not be suitable for monitoring the progression of IPF in patients with an emphysema extent of 15% [5]. At the same time, the diminished diffusing capacity may be attributable to the overlapping negative effects of both decreased capillary blood volume in emphysema and alveolar membrane thickening from pulmonary fibrosis, resulting in more significant reductions in DLCO [10]. Thus, if only spirometry is performed, the relative preservation of spirometric values may lead to an underdiagnosis of chronic lung disease. 

The blood gas analysis of CPFE differs from that of COPD and resembles that of IPF more closely. Severe decreases in arterial oxygen saturation and hypoxemia during exercise are highly prevalent in CPFE, particularly when complicated by severe PH [51,54]. Therefore, exercise limitation accompanied by a decrease in oxygen saturation and an isolated and/or severe decrease in DLCO or KCO, in contrast to a mild ventilatory defect, should raise the suspicion of CPFE and/or PH. Hypercapnia occurs very late in the progression of the disease. The functional profile is similar when CPFE is present in CTD or fHP [16,24].

## 8. Imaging

Imaging demonstrates a variety of findings on chest high-resolution computed tomography (HRCT) dominated by the coexistence of emphysema in the upper zones of the lungs and pulmonary fibrosis in the lower zones [2]. The fibrotic lesions are heterogeneous and commonly represented by the UIP pattern with honeycomb images in 95% of cases, subpleural reticular opacities in 87%, and traction bronchiectasis in 73% of cases [2]. Emphysematous lesions in patients with CPFE include diffuse (centrilobular and/or bullous) emphysema (Figure 2) or paraseptal emphysema with subpleural predominance [2]. In an earlier publication, Cottin et al. suggested that CPFE patients typically have predominantly paraseptal emphysema, which is considered a distinctive feature of this syndrome, compared to the typical centrilobular smoking-related emphysema seen in COPD. However, there are currently no studies that have directly compared patterns of emphysema in CPFE and COPD [49]. The areas of fibrosis and emphysema may be completely separated, or they may transition gradually. Moreover, paraseptal emphysematous lesions may exist at the lung bases within the fibrotic lesions (Figure 3) [55].

Besides the UIP pattern, several other patterns have been reported, such as non-specific interstitial pneumonia (NSIP) [9]; respiratory bronchiolitis-associated interstitial lung disease (RB-ILD), represented by poorly defined centrilobular nodules [2]; desquamative interstitial pneumonia (DIP) showing ground glass opacities; and smoking-related interstitial fibrosis (SRIF) (Figure 4) [12]. As the risk of lung cancer appears to be higher in CPFE than in IPF or COPD alone [56,57], patients with CPFE may develop lung nodules or masses.

A distinctive imaging feature of CPFE that does not appear to be present in patients with IPF or emphysema alone is the thick-walled large cyst pattern resulting from the combination of emphysema and SRIF [6], i.e., development of pulmonary fibrosis in the emphysematous lung [55]. These cysts are larger than honeycombing cysts (>1 cm in diameter) and have a 1 mm wall thickness. They may be located in the upper lungs in subpleural areas or develop in the reticulation and/or honeycombing regions (Figure 2) [7]. 

It has been found that the presence of these large cystic lesions with thick walls in CPFE patients suggests greater severity of emphysema compared to patients with no such lesions [58].

Both clinicians and radiologists should remember that HRCT is an important prognostic tool since the extent of fibrosis has a more significant impact than emphysema [59], and pulmonary fibrotic changes may be more important contributors to disease progression than emphysema [60]. 

## 9. Comorbidities

### 9.1. Lung Cancer

As smoking-related diseases, emphysema and IPF have been identified as independent risk factors for developing lung cancer [61,62].

CPFE patients appears to have a higher incidence of lung cancer (6.1–46.8%) [53] compared to IPF (7–20%) [62] or COPD (12–14%) [63]. The study published by Yoo and colleagues on IPF patients with lung cancer revealed that male gender, smoking at the time of IPF diagnosis, and an annual decline of 10% or more in FVC were risk factors for lung cancer [64]. Similarly, lung cancer in CPFE was typically diagnosed in elderly, heavy smokers who are predominately male with a median survival time of 19.5 months [65]. Another study revealed that lung cancer in CPFE more frequently has a squamous cell carcinoma histology and a lower lobe predominance occurring in the field of fibrosis, it is diagnosed in the advanced pathological stage, and is associated with increased mortality [66]. In addition, CPFE patients with concomitant non-small cell lung cancer (NSCLC) are at a greater risk for acute exacerbation than IPF patients with NSCLC [67], and thus have an increased mortality [53]. 

### 9.2. Pulmonary Hypertension

Pulmonary hypertension (PH) is a significant complication of COPD, IPF, and CPFE and is associated with a lower survival rate [68]. Approximately 50% of COPD cases, 31–46% of advanced IPF cases [68,69], and 15%–55% of CPFE patients are complicated by PH [6]. 

While PH in COPD or IPF patients is generally mild to moderate, in most CPFE cases, it is moderate to severe [69]. This severity is likely due to the combined effects of emphysema and fibrosis, leading to a reduction in the pulmonary capillary bed and hypoxic vascular constriction, resulting in a more severe increase in pulmonary vascular resistance. Furthermore, chronic inflammation induced by cigarette smoke in susceptible individuals may contribute to vascular remodeling [9]. Pulmonary artery intimal fibrosis and medial hypertrophy, thickening, and fibrosis of the pulmonary veins are among the changes that occur, leading to significant luminal obstruction and modest venopathy, but no apparent capillary alterations [70]. Interestingly, in CPFE-induced PH, narrowing of pulmonary arteries due to muscular layer hypertrophy is a global finding, even in areas of normal lung tissue [70]. 

PH becomes a dominant clinical feature and a major complication of CPFE, resulting in severe dyspnea, markedly impaired gas transfer (DLCO), and exercise hypoxemia, all of which contribute to a poor prognosis [6]. Given the added mechanisms contributing to increased pulmonary pressure, patients with CPFE and PH have a lower survival rate compared to patients with PH and IPF (25 vs. 34 months) [71] or COPD (5-year survival rate 25% vs. 36%) [68].

## 10. Treatment

Currently, there is limited evidence regarding CPFE treatment; therefore, management strategies are based solely on data derived from studies on COPD and IPF patients with concurrent CPFE. Encouraging general measures such as smoking cessation and avoiding environmental exposures may slow down disease progression [17,72]. Because infectious exacerbations significantly deteriorate lung function, vaccination against influenza viruses, COVID-19, and *Streptococcus pneumoniae* is considered beneficial [6]. 

Although inhaled bronchodilators with or without corticosteroids have shown to be effective in CPFE patients with significant airflow limitation [73,74], the evidence is scarce, and further studies are needed to address patients with preserved lung function.

Currently, only two drugs, pirfenidone and nintedanib, have proven efficacy in both IPF and fibrotic ILD, reducing the progression rate by half [75,76,77]. However, in patients with IPF who have emphysema, FVC decline, which is a traditional primary endpoint in most trials, is attenuated, and there is a lack of clear-cut efficacy in this group of patients. Despite this, the ATS/ERS/JRS/ALAT guideline suggests that antifibrotic medication may be beneficial for IPF patients with CPFE [6]. Similarly, patients with CTD-ILD or fibrotic HP and emphysema may benefit from therapy with glucocorticoids and/or immunosuppressive agents [78]. 

Regarding therapy for pulmonary hypertension, there is a significant discrepancy between controlled and uncontrolled data. As a result, there is limited evidence regarding the use of pulmonary vasodilators, leading to limited treatment options [79]. However, uncontrolled observational studies suggest that sildenafil may be beneficial for patients with IPF [80,81], and the current clinical guideline states that in patients with severe PH associated with ILD, phosphodiesterase 5-inhibitors may be considered, based on individual decision making [82]. In CPFE patients with hypoxemia at rest or during exercise, oxygen supplementation may alleviate pulmonary hypertension [74]. 

The approach to treating lung cancer in CPFE patients is similar to other patients. Unfortunately, the combined severity of emphysema and underlying fILD likely contribute to the greater complication rates among CPFE individuals receiving treatment [83]. Moreover, studies suggest a higher rate of exacerbation and malignancy recurrence compared to patients without CPFE, which increases the postsurgical risk of morbidity and mortality (Table 2).

## 11. Conclusions

CPFE is a distinct entity that sums up the pathogenic pathways found in both COPD and IPF, which determine the impaired regeneration of the lung parenchyma after damage. More research is needed to understand the coexistence of the divergent phenotypes of response to injury driven by environmental factors in CPFE. This entity has a wide variety of imaging and histopathological appearances. From the clinical perspective, CPFE combines the effects of emphysema and fibrosis, resulting in patients with increased symptoms that frequently associate with severe comorbidities such as lung cancer and pulmonary hypertension, which impose a poor prognosis and increased mortality. Treatment options are scarce, as they derive from studies on COPD and IPF patients with concurrent CPFE, and include similar recommendations with limited supporting evidence. 

## Figures and Tables

**Figure 1 biomedicines-11-01636-f001:**
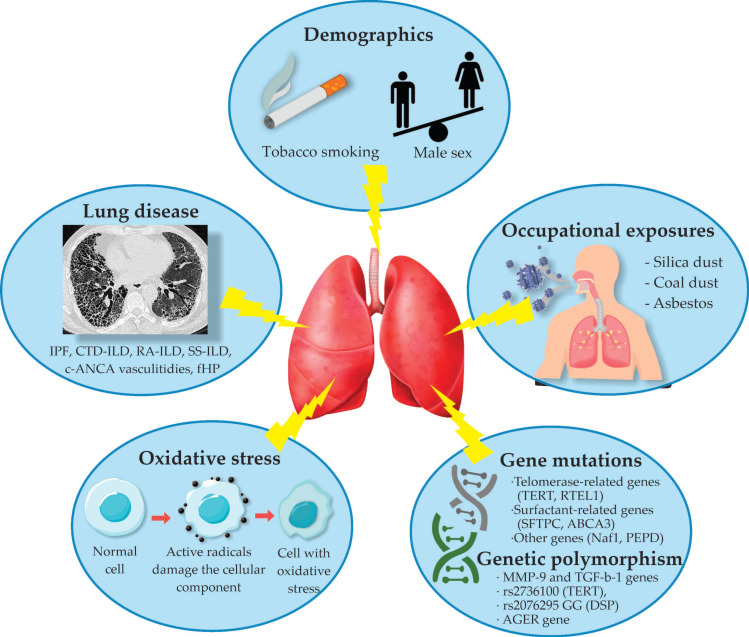
Factors associated with Combined Pulmonary Fibrosis and Emphysema. Abbreviations: IPF—idiopathic pulmonary fibrosis, CTD-ILD—connective tissue disease-associated interstitial lung disease, RA-ILD—rheumatoid arthritis-associated interstitial lung disease, SS-ILD—systemic sclerosis-associated interstitial lung disease, c-ANCA—antineutrophil cytoplasmic autoantibody, fHP—fibrotic hypersensitivity pneumonitis.

**Figure 2 biomedicines-11-01636-f002:**
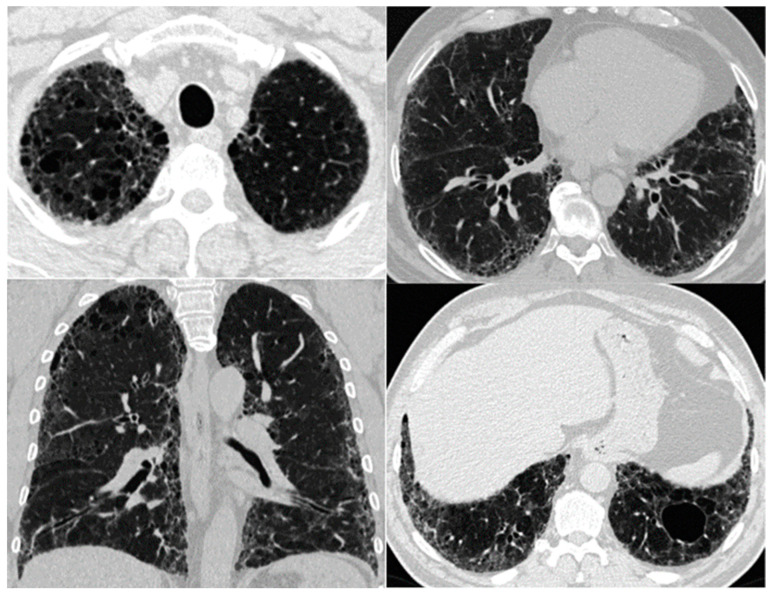
HRCT patterns in CPFE. A predominant pattern of centrilobular emphysema in the upper lobes and a thick-walled cyst in the lower left lung zone in a field of fibrotic lesions in a 60-year-old patient diagnosed with idiopathic pulmonary fibrosis.

**Figure 3 biomedicines-11-01636-f003:**
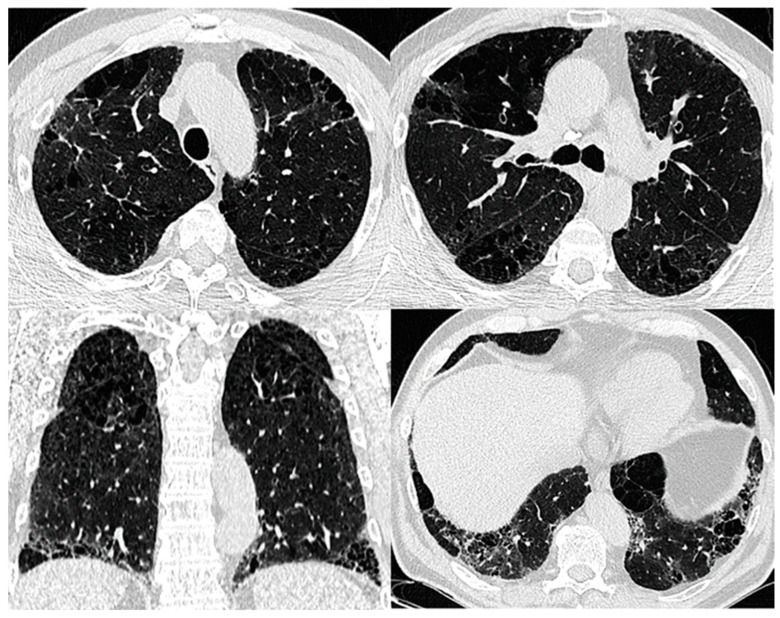
Progressive transition HRCT pattern of CPFE. The typical distribution: predominant pattern of centrilobular emphysema in the upper lobes, extending to a panlobular emphysema in the midzones, and an isolated area of centrilobular emphysema in the left lower lobe in the field of subpleural reticular opacities, traction bronchiectasis, and bronchioloectasis in a 59-year-old patient with fibrotic hypersensitivity pneumonitis with a history of 30 pack-years of smoking.

**Figure 4 biomedicines-11-01636-f004:**
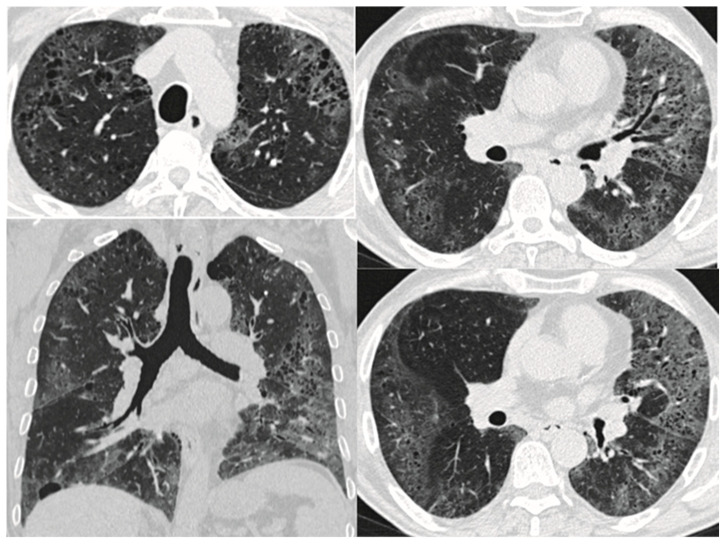
HRCT pattern of emphysema admixed with DIP. Areas of low attenuation (emphysema) are mixed with ground-glass opacities (areas of high attenuation) and periemphysematous thickening.

**Table 1 biomedicines-11-01636-t001:** Distinctive functional features of CPFE compared to its separate components.

Pulmonary Function Test Variable	CPFE	Fibrotic ILD	Emphysema
FVC	N/↓	↓	N/↓
FEV1	N/↓	↓	N/↓
FEV1/FVC	N/↓/↑	N/↑	N/↓
TLC	N/↓/↑	N/↓	↑
FRC	N/↓/↑	N/↓	↑
RV	N/↓/↑	N/↓	↑
DLCO	↓ disproportionately	↓	↓
KCO	↓↓	N/↓	↓
SaO_2_ following 6 min walk test	↓↓	↓	↓

Abbreviations: FVC—forced vital capacity, FEV1—forced expiratory volume in the 1st second, FEV1/FVC—Tiffeneau index, TLC—total lung capacity, FRC—functional residual capacity, RV—residual volume, DLCO—diffusion lung capacity for carbon monoxide, KCO—carbon monoxide transfer coefficient, SaO_2_—oxygen saturation, N—normal, ↓—decreased, ↓↓—very decreased, ↑—increased.

**Table 2 biomedicines-11-01636-t002:** Comparative features of CPFE with COPD and IPF.

	COPD	CPFE	IPF
**Prevalence**	↑↑↑	↓↓	↓
**Smoking history**	↑↑↑	↑↑↑/−	↑
**Gender predilection**	male predominance	male predominance	male predominance
**Age of clinical manifestations**	5th–6th decade	6th–7th decade/earlier in CTD-ILD	5th–6th decade
**Spectrum of environmental exposures**	Tobacco, air pollution	Tobacco, noxious particles, or gases other than tobacco, asbestos, silica dust, and agrochemical chemicals, organic dust	Organic dust, metal and mineral dust, wood dust, asbestos, and ambient particulate matter
**Autoimmunity**	−	−/+	−
**Response to oxidative stress**	cellular injury and death and ECM ↓	cellular injury and death and ECM ↓ in the upper lobes, Activation of fibroblasts, ECM ↑ in the lower lobes	Activation of fibroblasts, ECM ↑
**Role of epigenetic factors**			
- FAM13A allele	↓	↑	↑
- AGER gene rs2070600	↑	↑	↓
- T allele of MMP9	↓	↑	↑
- rs2736100 C allele of TERT	−	↓	↓
**Pathology**	Small airways remodeling, fibrosis, and destruction of the lung parenchyma with an upper lobe predominance	Emphysema in the upper lobes and patchy fibrosis, fibroblast foci, and honeycombing—UIP/NSIP/DIP pattern—in the lower lobes. Presence of thick-walled cysts on a UIP background	Lung parenchyma and interstitium, with a lower lobe predilection
**Clinical features**			
- Symptoms	Cough with sputum production, dyspnea later in the disease course	Dyspnea, dry cough later in the disease course	Dyspnea, dry cough later in the disease course
- “Velcro” sounds	−	+	+
- Finger “clubbing”	Not a typical sign	50%	70%
**Lung function**	Obstruction +/− Low DLCO	Normal Low DLCO	Restriction Low DLCO
**Imaging**	Centrilobular and/or bullous emphysema	Coexistence of paraseptal emphysema in the upper zones of the lungs and the UIP/NSIP/DIP in the lower zones, thick-walled cysts in the area of fibrosis	UIP pattern
**Comorbidities**			
- Lung cancer risk	↑	↑↑	↑
- Pulmonary hypertension	↑	↑↑	↑
**Treatment options**	Bronchodilators +/− ICS +/− Oxygen therapy	Bronchodilators +/− ICS Antifibrotics (pirfenidone, nintedanib) Oxygen therapy	Antifibrotics (pirfenidone, nintedanib) Oxygen therapy

Abbreviations: ↑increased, ↑↑ - moderately increased, ↑↑↑ - very increased ↓—decreased, ↓↓ - moderately decreased, “+”—present, “−”—absent, ECM—extracellular matrix, UIP—usual interstitial pneumonitis, NSIP—nonspecific interstitial pneumonitis, DIP—desquamative interstitial pneumonitis, ICS—inhaled corticosteroids.

## Data Availability

Not applicable.

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
