# Peer review of "Combined Pulmonary Fibrosis and Emphysema: Comparative Evidence on a Complex Condition"

_biomedicines, 2023, doi:10.3390/biomedicines11061636_

Round 1
Reviewer 1 Report
The authors summarized the knowledge regarding combined pulmonary fibrosis and emphysema based on presently available literature. This distinct condition carries the clinical features of emphysema and pulmonary fibrosis with the dominance of clinical symptoms of fibrosis. The more severe clinical outcome of this form of pulmonary disease compared to COPD or forms of pulmonary fibrosis signifies the importance of this manuscript. Furthermore, the limited interventional tools to treat the condition point out the necessity of early diagnosis and awareness of clinical picture. The manuscript discusses all major aspects of the disease, from diagnosis to treatment, with a summary of genetic predisposition. The references are appropriate and support the review.
Comments:
1) Please move Figure 1 to the left. Part of the figure was cut off.
2) Please rephrase the sentence in the section “Clinical features” at lines 156-157. “In almost all patients, progressive shortness of breath is the most frequent and is frequent and classic symptom and is typically more severe, especially during physical exertion.” This sentence is confusing.
3) Please rephrase the sentence at lines 158-159. Exertional dyspnea is not a limiting factor, but rather a clinical sign/symptom of inefficient ventilation.
4) Please delete one of the "is" at line 277.
The English language is good. Some sentences need to be rephrased (see comments).
Author Response
Dear reviewer,
thank you for your valuable input to our manuscript. We have addressed all the objections, as follows:
1) Please move Figure 1 to the left. Part of the figure was cut off – the figure was moved to the left and arranged according to the MDPI template.
2) Please rephrase the sentence in the section “Clinical features” at lines 156-157. “In almost all patients, progressive shortness of breath is the most frequent and is frequent and classic symptom and is typically more severe, especially during physical exertion.” This sentence is confusing. –We apologize for this incoherent phrase. The sentence has been rephrased to “In almost all patients, progressive shortness of breath is the most common symptom and is typically more severe, especially during physical exertion.”
3) Please rephrase the sentence at lines 158-159. Exertional dyspnea is not a limiting factor, but rather a clinical sign/symptom of inefficient ventilation. - Thank you for your suggestion. The sentence has been rephrased to “Exertional dyspnea is the clinical expression of inefficient ventilation and increased dead space ventilation in hypoperfused areas”.
4) Please delete one of the "is" at line 277. – According to your objection one of the "is" has been deleted.
Reviewer 2 Report
In the present review Calaras and colleagues aimed to review the recent insights on combined pulmonary fibrosis and emphysema (CPFE). The authors provide an overview on the main clinicopathological characteristic defining the CPFE and provide parallelism with other chronic pulmonary disease such as COPD and idiopathic pulmonary fibrosis (IPF). They also discussed about the diagnostic value of high-resolution computed tomography as well as provided a hint on the current treatment.
· First of all, I suggest the authors to change the title, which did not reflect the main evidence come from the review. Rather than novel insights, it would be useful refer to comparative evidence describing a complex condition.
· I suggest the authors to re-edit the paper by focusing it on the comparison between CPFE with COPD and IPF (or other similar pulmonary diseases), as they have partially done in the present review. Given CPFE is a recently recognized syndrome this approach could be more effective and informative than reporting evidence already discussed in the literature. Moreover, in my opinion, a more detailed focus on the analogies and differences in the molecular mechanisms driving fibrosis, inflammation and oxidative stress must be provided and discussed in appropriate sections.
· In introduction paragraph, the authors should provide and in-depth characterization of the pathological features of COPD and IPF (see. PMIS 35326114; 37181377), to facilitate the reading of the other sections in which the comparisons appear.
· A more recent literature (within five years) must be used to update the prevalence data reported in section 3 (lanes 47-57)
· Figure 2, 3 and 4: the authors must provide the sources of the HRCT images used. This information must be reported in the text or in the legends
· The authors must provide a conclusions paragraph, in which they have to summarize the main evidence arose from the evidence argued in the review. In my opinion the authors should provide a final image and/or a table in which the analogies and differences among CPFE and the other pulmonary diseases considered in the text would be summarized.
There are not issues
Author Response
First of all, on behalf of all the authors I would like to thank the reviewer for their suggestions that have improved the quality of the manuscript.
Please find below point-by-point response your comments:
- First of all, I suggest the authors to change the title, which did not reflect the main evidence come from the review. Rather than novel insights, it would be useful refer to comparative evidence describing a complex condition.
Response: Thank you for this insightful suggestion. The title has been changed to “Combined pulmonary fibrosis and emphysema: comparative evidence on a complex condition”.
- I suggest the authors to re-edit the paper by focusing it on the comparison between CPFE with COPD and IPF (or other similar pulmonary diseases), as they have partially done in the present review. Given CPFE is a recently recognized syndrome this approach could be more effective and informative than reporting evidence already discussed in the literature. Moreover, in my opinion, a more detailed focus on the analogies and differences in the molecular mechanisms driving fibrosis, inflammation and oxidative stress must be provided and discussed in appropriate sections.
Response: Thank you for your valuable suggestions. As the reviewer has kindly stressed, that the manuscript shows a comparison between features of CPFE, COPD, and IPF, we have focused more on this strategy and supplemented the pathogenesis with additional data.
- In introduction paragraph, the authors should provide and in-depth characterization of the pathological features of COPD and IPF (see. PMIS 35326114; 37181377), to facilitate the reading of the other sections in which the comparisons appear
Response: Thank you for this suggestion. However, the pathological features of COPD and IPF are well established, described in numerous textbooks and are beyond the scope of our work that is a focused update on CPFE. We have added a short summary of the main characteristics of all three disease entities to help the reader, but an “in-depth” description is beyond the scope of our work and we would like to kindly decline further extending the description of pathological features of COPD and/or IPF.
- A more recent literature (within five years) must be used to update the prevalence data reported in section 3 (lanes 47-57)
Response: We have updated the prevalence data with more recent literature as suggested (reference No 3).
- Figure 2, 3 and 4: the authors must provide the sources of the HRCT images used. This information must be reported in the text or in the legends.
Response: This issue has been addressed as suggested. We provided the information regarding the source in the legends marked with a * and at the end of the manuscript.
- The authors must provide a conclusions paragraph, in which they have to summarize the main evidence arose from the evidence argued in the review. In my opinion the authors should provide a final image and/or a table in which the analogies and differences among CPFE and the other pulmonary diseases considered in the text would be summarized.
Response: The conclusions were added and we have added a summarizing table as well.
Round 2
Reviewer 2 Report
Althought the authors did not provide all the insights required, the manuscript could be suitable for publication.
MINOR ISSUES:
Please modify the sentence :"and comparative evidence describing features of its components – COPD and IPF". Do COPD and IPF are CPFE components? Did the authors mean "...comparative evidence describing features of its components and the analogies and differences with two common pulmunary disease as COPD and IPF" ?
Line 392, please provide the affiliation of dr. Caralas
I've not suggestions
Author Response
Dear reviewer,
Please allow me to express our gratitude for your valuable input on behalf of all authors.
Please find below the responses to the additional objections:
- Please modify the sentence:"and comparative evidence describing features of its components – COPD and IPF". Do COPD and IPF are CPFE components? Did the authors mean "...comparative evidence describing features of its components and the analogies and differences with two common pulmunary disease as COPD and IPF" ?
RESPONSE: Thank you for this objection. In order to avoid misconceptions we have change the phrase to: „comparative evidence describing COPD and IPF”
- Line 392, please provide the affiliation of dr. Caralas
RESPONSE: Thank you for this suggestion. We have added the affiliation in the indicated line. Moreover, since dr. Calaras is the first author of the manuscript, the affiliation is also provided in the corresponding compartment (line 5).